# No Association between Glucocorticoid Diurnal Rhythm in Breastmilk and Infant Body Composition at 3 Months

**DOI:** 10.3390/nu11102351

**Published:** 2019-10-02

**Authors:** Jonneke Hollanders, Lisette R. Dijkstra, Bibian van der Voorn, Stefanie M.P. Kouwenhoven, Alyssa A. Toorop, Johannes B. van Goudoever, Joost Rotteveel, Martijn J.J. Finken

**Affiliations:** 1Emma Children’s Hospital, Amsterdam UMC, Pediatric Endocrinology, Vrije Universiteit Amsterdam, 1000-1183 Amsterdam, The Netherlands; l.dijkstra@amsterdamumc.nl (L.R.D.); a.toorop@amsterdamumc.nl (A.A.T.); j.rotteveel@amsterdamumc.nl (J.R.);; 2Department of Paediatric Endocrinology, Obesity Center CGG, Sophia Children’s Hospital, 3000-3099 Rotterdam, The Netherlands; b.vandervoorn@erasmusmc.nl; 3Emma Children’s Hospital, Amsterdam UMC, Department of Pediatrics, Vrije Universiteit Amsterdam, 1000-1183 Amsterdam, The Netherlandsh.vangoudoever@amsterdamumc.nl (J.B.v.G.)

**Keywords:** cortisol, cortisone, growth, circadian rhythm, human milk

## Abstract

Objective: Glucocorticoids (GCs) in breastmilk have previously been associated with infant body growth and body composition. However, the diurnal rhythm of breastmilk GCs was not taken into account, and we therefore aimed to assess the associations between breastmilk GC rhythmicity at 1 month and growth and body composition at 3 months in infants. Methods: At 1 month postpartum, breastmilk GCs were collected over a 24-h period and analyzed by LC-MS/MS. Body composition was measured using air-displacement plethysmography at 3 months. Length and weight were collected at 1, 2, and 3 months. Results: In total, 42 healthy mother–infant pairs were included. No associations were found between breastmilk GC rhythmicity (area-under-the-curve increase and ground, maximum, and delta) and infant growth trajectories or body composition (fat and fat free mass index, fat%) at 3 months. Conclusions: This study did not find an association between breastmilk GC rhythmicity at 1 month and infant’s growth or body composition at 3 months. Therefore, this study suggests that previous observations linking breastmilk cortisol to changes in infant weight might be flawed by the lack of serial cortisol measurements and detailed information on body composition.

## 1. Introduction

Growing attention is focused on the etiology of obesity, and it has been hypothesized that part of its origin can be traced back to events occurring in early life (i.e., the Developmental Origins of Health and Disease (DOHaD) hypothesis) [1].

Given its effects on fat disposition and metabolism, the hypothalamus-pituitary-adrenal (HPA) axis has been implicated to play a role in the pathway leading to obesity [2,3]. Not only endogenous but also maternal glucocorticoids (GCs) appear to be involved. Evidence from animal experiments indicates that increased transplacental supply of maternal GCs may be associated with a lower birth weight and cardiovascular correlates, such as hypertension and hyperglycemia [4]. In humans, fetal exposure to excess maternal cortisol, e.g., due to maternal anxiety or depression, has been associated with a higher risk of childhood adiposity [5].

After birth, small amounts of maternal GCs appear to be transferred to the developing infant through breastmilk. Maternal GCs in breastmilk have been shown to cross the intestinal barrier in animals [6], and have been associated with growth and body composition. Hinde et al. (2015) [7] found that cortisol in the breastmilk of rhesus macaques was positively associated with weight gain in offspring. In humans, Hahn-Holbrook et al. (2016) [8] showed that cortisol in breastmilk at the age of 3 months was inversely associated with body mass index (BMI) percentile gains in the first 2 years of life. Whether the findings from these studies are contradictory is unclear, since length was not taken into account by Hinde et al. [7]. Moreover, the effect of GCs on growth might change between the ages of 3 months and 2 years.

Our group has previously shown that GCs in breastmilk follow maternal HPA-axis activity, with a peak in the morning and a nadir at night [9]. Although previous studies have found associations between cortisol in breastmilk and growth of offspring, none of them took GC rhythmicity into account. However, obesity has previously been associated with a flatter diurnal cortisol slope in adults [10], and there is also some evidence that a blunted GC rhythm is associated with obesity in children [11,12]. Both Hinde et al. (2015) [7] and Hahn-Holbrook et al. (2016) [8] did not collect samples around peak GC levels, while Hahn-Holbrook et al. (2016) also had a wide time window during which samples could be collected (11:30–16:00). 

We therefore aimed to assess the associations between breastmilk GC rhythmicity and infant growth and body composition. We measured cortisol and cortisone in breastmilk at 1 month of age over a 24-hour period, measured body composition using air-displacement plethysmography at 3 months of age, and collected length and weight data monthly up to that age. Due to associations found between a blunted endogenous GC rhythm and obesity in both children and adults [10,11,12], we hypothesized that less GC variability in breastmilk could be associated with a higher fat mass in the infants.

## 2. Materials and Methods 

### 2.1. Population

Healthy mother–infant pairs were recruited at the maternity ward of the Amsterdam UMC, location VUMC (a tertiary hospital) in the Netherlands between March 2016 and July 2017. Subjects were eligible for inclusion when infants were born at term age (37–42 weeks) with a normal birth weight (−2 to +2 SDS), and when mothers had the intention to breastfeed for a minimum of three months. Exclusion criteria were: (1) Major congenital anomalies, (2) multiple pregnancy, (3) pre-eclampsia or HELLP, (4) medication use other than “over the counter” drugs, (5) maternal alcohol consumption of >7 IU/week, and/or 6) a maternal temperature of >38.5 °C at the time of sampling. Approval of the Medical Ethics Committee of the VUMC was obtained (protocol number 2015.524), and written informed consent was obtained from all participating mothers.

### 2.2. Data Collection

#### 2.2.1. Peripartum

Shortly after inclusion, within the first days postpartum, mothers filled in a questionnaire pertaining to their pregnancy and birth, as well as maternal and infant anthropometric and demographic data. 

#### 2.2.2. One Month Postpartum

At 30 days postpartum (±5 days), mothers collected a portion of breastmilk (1–2 mL) prior to each feeding moment, over a 24-h period (i.e., five to eight times). Although only foremilk was collected through this method, previous research has shown that GC concentrations are similar in fore- and hindmilk [13]. Mothers could follow their own feeding schedule and were therefore asked to report the exact time of sampling. Milk was collected manually or with a breast pump; we requested that mothers used the same method for all samples. Milk was stored in the mother’s freezer, and subsequently in the laboratory at −20 °C for less than 3 months prior to analysis.

At the time of sampling, maternal distress was quantified with the Hospital Anxiety and Depression Scale (HADS) [14]. This questionnaire contains 14 questions scored from 0 to 3, which assess self-reported levels of depression and anxiety symptoms. Seven questions concern depressive symptoms (HDS) and seven questions assess anxiety symptoms (HAS). A score of ≥8 on a subscale is indicative of clinically relevant depression and/or anxiety symptoms.

#### 2.2.3. Three Months Postpartum

At 3 months postpartum (±2 weeks), the body composition of the infants was assessed with the Pea Pod, an air-displacement plethysmography (ADP) system (COSMED USA, Inc., Concord, CA, USA) [15]. It is based on a bi-compartmental model, which uses pressure and volume changes in the chamber through which body density was determined. Age- and sex-specific fat and fat free mass density values were subsequently used to calculate fat mass (FM) and fat free mass (FFM) [15]. 

As part of the national standard care, weight and length at 1, 2, and 3 months of age were measured by the staff of the child health clinic and were obtained through a questionnaire. Weight was measured fully undressed on a balance scale with an accuracy of 1 g. Length was measured in the supine position to the nearest 0.1 cm. Additionally, all mothers were asked if their infants were still breastfed for >80% at the age of 3 months.

### 2.3. Laboratory Analysis

Cortisol and cortisone concentrations in breastmilk were determined by isotope dilution liquid chromatography–tandem mass spectrometry (LC–MS/MS), as previously described [16]. In brief, internal standards (13C3-labeled cortisol and 13C3-labeled cortisone) were added to 200 μL of the samples. Then, breastmilk was washed 3 times with 2 mL of hexane to remove lipids. Finally, samples were extracted and analyzed using Isolute plates (Biotage, Uppsala, Sweden) and analyzed by LC-MS/MS (Acquity with Quattro Premier XE, Milford MA, USA, Waters Corporation). The intra-assay coefficients of variation (CV%) were 4 and 5% for cortisol levels of 7 and 23 nmol/L, and 5% for cortisone levels of 8 and 33 nmol/L for LC-MS/MS measurements. The inter-assay CV% was <9% for both cortisol and cortisone. The lower limit of quantitation (LLOQ) was 0.5 nmol/L for both cortisol and cortisone. All samples were measured in duplicate.

### 2.4. Statistics

First, data of GC concentrations in breastmilk were converted into the following rhythm parameters, in order to provide a full overview of GC rhythmicity:The maximum GC concentration, as a proxy for peak concentrations;The delta between maximum and minimum GC concentrations, as a measure of rhythm variability; andArea under the curve (AUC) ground (g) and increase (i), using the trapezoid rule [17].Calculations were corrected for total sampling time, since this differed between mothers. AUCg is a measure of total GC exposure, while AUCi provides information on GC variability.

Mother–infant pairs were excluded from analyses when no valid GC data was available around the time of the expected morning peak (5:00–10:00) and/or when the total sample collection was <8 h.

Fat% was determined from the FM and FFM values. The fat mass index (FMI) and fat free mass index (FFMI) were calculated by dividing FM and FFM values (in kg), respectively, by infant length squared (m^2^), since fat mass and fat free mass are known to change with length [18]. Length and weight data were converted to SDS [19,20]. Body mass index (BMI) was only calculated at 3 months of age, and converted to SDS [19].

Linear regressions were used to assess the associations between GC rhythm parameters at 1 month of age and length SDS, weight SDS, BMI SDS, FMI, and FFMI at 3 months of age. First, unadjusted regression analyses were performed. Next, the following potential confounders were tested: Sex, HADS-score (HAS and/or HDS ≥ 8), pre-pregnancy BMI, ethnicity (Caucasian vs. non-Caucasian), socio-economic status, birth weight SDS, gestational age, weight gain during pregnancy, parity (1 vs. >1), mode of delivery (vaginal vs. caesarian section), and % breastmilk at 3 months of age (< or >80%). Due to our sample size, the three variables with the largest confounding effect (i.e., largest change in β of the independent variable) were used for the multiple linear regression analyses. Thus, weight gain during pregnancy, % breastmilk at 3 months of age, and ethnicity were included in the final model assessing the association between GC rhythm parameters and body composition outcomes. No effect modification was found for infant sex, and analyses were therefore not stratified.

Lastly, length and weight SDS growth trajectories between 1 and 3 months of age were plotted against AUCi and AUCg values by using generalized estimating equations (GEEs), and 95% confidence intervals were calculated according to the method described by Figueiras et al. [21]. AUCi and AUCg outcomes for cortisol and cortisone were categorized as ≤p25, p25–75, and ≥p75.3. 

## 3. Results

### 3.1. Population

Forty-four mother–infant pairs were included in the study. One mother–infant pair was lost to follow-up, three mother–infant pairs returned the growth questionnaires but did not consent to the Pea Pod measurement, and one pair was excluded because no samples were collected between 5:00 and 10:00 and/or because the total sampling time was <8 h. Therefore, a total of 42 mother–infant pairs were included in the growth trajectory analyses, whereas 39 mother-infant pairs were included in the body composition analyses at 3 months of age. Of the included mother–infant pairs, 59.5% were mother–son pairs. Table 1 shows the characteristics of the population. Appendix A shows the cortisol and cortisone concentrations in breastmilk in 4-h intervals.

### 3.2. Linear Regression Analyses

No associations were found between the GC rhythm parameters (AUCi, AUCg, maximum, and delta) and body composition in the unadjusted analyses. Adjusting the analyses for weight gain during pregnancy, % breastmilk at 3 months of age, and ethnicity did not change the results (Table 2).

### 3.3. Growth Trajectories

Figure 1 shows the growth trajectories for length and weight SDS according to breastmilk cortisone AUCi and AUCg outcomes. No differences were found between the categories ≤p25, p25–75, and ≥p75. Results for breastmilk cortisol AUCi and AUCg were similar (data not shown).

## 4. Discussion

In this study, despite increased evidence for associations between blunted endogenous GC rhythms and obesity in both children and adults [10,11,12], no associations were found between GC rhythmicity in breastmilk sampled at 1 month and infant body composition or growth at 3 months. Therefore, our study could not confirm previous observations in animals and humans. Hinde et al. (2015) [7] measured cortisol in breastmilk of 108 rhesus macaques at 1 month of age, and analyzed growth outcomes at 3.5 months of age. They found that higher cortisol concentrations were associated with greater weight gain over time. Hahn-Holbrook et al. (2016) [8] studied associations between breast-milk cortisol and BMI gains up until the age of 2 years in 51 mother-infant pairs. They found that higher milk cortisol concentrations were associated with smaller BMI gains in offspring.

The different results between this study and previous studies could have several explanations. First, cortisol sampling in the previous studies did not take the diurnal rhythm of breastmilk GCs into account. Hinde et al. [7] sampled between 11:30 and 13:00, which did not capture the peak GC concentration, since in Rhesus macaques, similar to humans, this occurs at around 8:00. Hahn-Holbrook et al. [8] collected a single breastmilk sample within a wide time window (11:30–16:00), which also did not capture peak concentrations. Analyses were corrected for time of collection, but it has previously been shown that correcting for the time of sampling cannot account for all the variability observed in cortisol levels [9,22]. Second, in our study, GC concentrations were determined by LC-MS/MS, which has been shown to be more sensitive and reliable than radioimmunoassay and chemiluminescent immunoassay [23], which were used by Hinde et al. and Hahn-Holbrook et al., respectively. Lastly, it has previously been shown that increases in fat mass are specifically associated with mid-childhood overweight and obesity [24]. Therefore, in this study, body composition was measured by ADP, which is able to differentiate between fat mass and fat free mass. In contrast, weight gain and changes in BMI were used as outcomes measured by Hinde et al. [7] and Hahn-Holbrook et al. [8], respectively, both of which are less precise methods to determine body composition. Our more detailed methods when measuring GC concentrations in breastmilk as well as when determining body composition might therefore have led to more accurate conclusions.

Alternatively, the absence of associations might be due to the small sample size in this study, especially compared to Hinde et al. [7], who included 108 mother–infant pairs, resulting in more power to detect small differences. However, this should be balanced against the use of air-displacement plethysmography in this study, which is superior to weight gain for the assessment of body composition. Additionally, our follow-up until the age of 3 months was rather short. In contrast, follow-up took place up to 2 years of age in Hahn-Holbrook et al.’s study [8]. It is therefore possible that effects of GCs in breastmilk might only be noticeable at a later age. On the other hand, an increasing number of nutritional, lifestyle, and family factors determine body composition with advancing age, and it is therefore progressively more difficult to determine to what extent breastmilk cortisol explains BMI gains.

This study has several strengths and limitations. This was the first study to assess the association between GC rhythmicity in breastmilk and body composition in offspring. Body composition and GC rhythmicity were analyzed in detail, respectively, by the use of ADP and by measuring both cortisol and cortisone in breastmilk using samples that were collected over a 24-h period. Cortisone concentrations have been shown to be more reliable than cortisol measurements, at least in saliva and hair [25,26]. This is possibly due to the local conversion of cortisol by 11β-hydroxysteroid dehydrogenase type 2, which leads to higher concentrations of cortisone [27]. However, this study also has its limitations. The sample size of this study was relatively small, and it is therefore possible that modest effects could not be detected. It was also not possible to correct for all potential confounders. However, many confounders were considered, and the three variables with the largest confounding effect were included in the final model, which did not change the results compared to unadjusted analyses. It is therefore unlikely that adjusting for more variables would have altered the results. Second, the follow-up in this study was relatively short, and it is therefore possible that breastmilk GC rhythmicity has an effect that is only noticeable at a later age. Additionally, selection bias cannot be ruled out, since we did not collect data on mothers who were eligible for inclusion but decided against participation and since we included mother–infant pairs at a (tertiary) hospital. The study population might therefore not reflect the general population; for example, 51% of the mothers gave birth via Caesarian section, compared to approximately 17% in the general population [28]. Lastly, the interplay between GCs and infant body composition is complex, and could be moderated by, for example, exposure to GCs and other conditions in utero, the number of feeds per day, and the extrauterine environment of the infants, including synchrony in mother–infant interactions as well as stressful events. Unfortunately, we were not able to take these factors into account.

## 5. Conclusions

This study did not find an association between breastmilk GC rhythmicity at 1 month of age and growth trajectories as well as body composition of the offspring at 3 months of age. Therefore, this study suggests that previous observations linking breastmilk cortisol to changes in infant weight might be flawed by the lack of serial cortisol measurements and detailed information on body composition.

## Figures and Tables

**Figure 1 nutrients-11-02351-f001:**
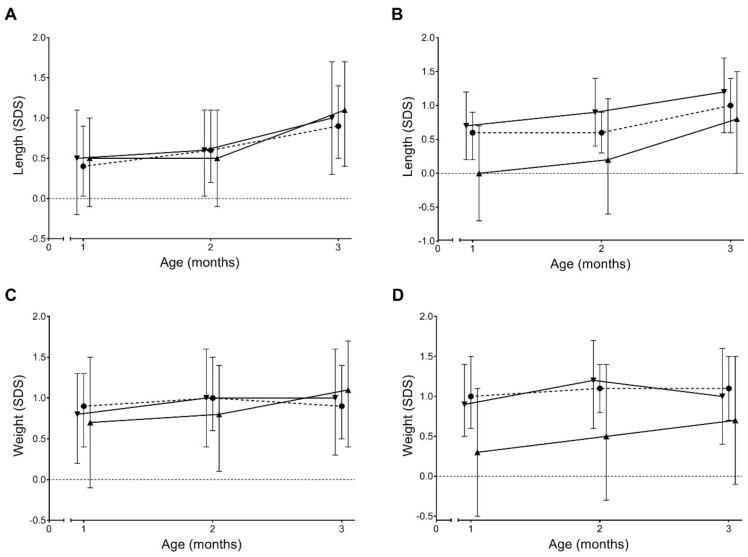
Growth trajectories between 1 and 3 months of age for length and weight, according to breastmilk cortisone AUC outcomes (*n* = 42). Results for breastmilk cortisol AUCi and AUCg were similar (data not shown). A = length for AUCi, B = weight for AUCg, C = weight for AUCi, D = weight for AUCg; ▼ = AUCi or g < p25, ● = AUCi or g = p25–75, ▲ = AUCi or g > p75, AUCi or g, area under the curve increase or ground.

**Table 1 nutrients-11-02351-t001:** Characteristics of the study population (*n* = 42).

	Unit	Mean ± SD or *n* (%)
Gestational Age	Weeks	39.9 ± 1.3
Birth weight	grams	3561 ± 498
	SDS	0.2 ± 1.0
Birth length *	cm	52.0 ± 2.6
	SDS	1.0 ± 1.6
Male sex		25 (59.5)
Primiparity		23 (54.8)
Caesarian section		21 (51.2)
HAS and/or HDS ≥ 8 at 1-month pp		6 (14.6)
Pre-pregnancy maternal BMI	kg/m^2^	22.3 ± 2.8
Weight gain during pregnancy	kg	13.1 ± 3.2
Maternal age	years	36.0 ± 4.7
Non-Caucasian ethnicity		8 (20.0)
Socioeconomic status	SDS	0.6 ± 1.2
>80% breastfed at 3 months of age		35 (87.5)
Age at breastmilk sampling	days	30.8 ± 2.6
Age at Pea Pod measurement **	days	90.5 ± 7.0

Values represent Mean ± SD or *n* (%); pp = postpartum. HAS, Hospital Anxiety Score; HDS, Hospital Depression Score. * *n* = 31, ** *n* = 39.

**Table 2 nutrients-11-02351-t002:** Adjusted associations between breastmilk glucocorticoid rhythmicity at 1 month of age and infant body composition at 3 months of age (*n* = 39).

		Length		Weight		BMI		FMI		FFMI		Fat %	
		B	95% CI	B	95% CI	B	95% CI	B	95% CI	B	95% CI	B	95% CI
**Cortisol**	Maximum	0.006	(−0.04 to 0.05)	0.022	(−0.02 to 0.07)	0.022	(−0.02 to 0.06)	0.003	(−0.04 to 0.05)	−0.006	(−0.04 to 0.03)	0.048	(−0.17 to 0.27)
	Delta	0.006	(−0.04 to 0.05)	0.024	(−0.02 to 0.07)	0.025	(−0.02 to 0.07)	0.003	(−0.04 to 0.05)	−0.004	(−0.04 to 0.03)	0.043	(−0.18 to 0.26)
	AUCi	0.025	(−0.15 to 0.20	0.101	(−0.07 to 0.28)	0.1	(−0.06 to 0.26)	0.06	(−0.10 to 0.23)	−0.095	(−0.23 to 0.04)	0.53	(−0.34 to 1.39)
	AUCg	0.029	(−0.13 to 0.19)	0.06	(−0.11 to 0.23)	0.046	(−0.11 to 0.20)	0.06	(−0.10 to 0.22)	−0.107	(−0.24 to 0.02)	0.53	(−0.28 to 1.33)
**Cortison**	Maximum	−0.002	(−0.04 to 0.03)	0.01	(−0.03 to 0.05)	0.014	(−0.02 to 0.05)	–0.006	(−0.04 to 0.03)	–0.006	(−0.04 to 0.02)	–0.007	(−0.19 to 0.18)
	Delta	−0.002	(−0.04 to 0.04)	0.018	(−0.02 to 0.06)	0.024	(−0.01 to 0.06)	–0.003	(−0.04 to 0.03)	–0.001	(−0.03 to 0.03)	0.005	(−0.19 to 0.20)
	AUCi	−0.005	(−0.09 to 0.08)	0.042	(−0.05 to 0.13)	0.055	(−0.02 to 0.14)	0.002	(−0.08 to 0.09)	–0.008	(−0.08 to 0.06)	0.034	(−0.41 to 0.48)
	AUCg	−0.001	(−0.07 to 0.07)	0.003	(−0.07 to 0.07)	0.004	(−0.06 to 0.07)	–0.013	(−0.08 to 0.05)	–0.02	(−0.08 to 0.04)	–0.02	(−0.37 to 0.33)

Values represent β (95% CI) as analyzed with linear regression. Analyses were adjusted for weight gain during pregnancy, % breastmilk at 3 months of age, and ethnicity. AUCi or g, area under the curve increase or ground; FMI, fat mass index; FFMI, fat free mass index.

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
