# Peer review of "No Association between Glucocorticoid Diurnal Rhythm in Breastmilk and Infant Body Composition at 3 Months"

_nutrients, 2019, doi:10.3390/nu11102351_

Round 1
Reviewer 1 Report
This manuscript concerns the study association between glucocorticoid diurnal rhythm in breastmilk and infant body composition at age 3 months. Authors obtained different results than previously published, so it is reasonable to further explore this topic. Authors present all the methodologies and results, discussed in details the limitation and advantages of the study and compare obtained results with other work. It would be interesting to compare these results with the data obtained for the infant fed with milk formula as they are not receiving glucocorticoids with milk.
1) Authors could extend the justification of the study in the introduction and discussion and explain the GC rhythmicity and its possible impact on developing an infant.
2) What was the amount of milk taken for the LC/MS-MS analysis? Was milk dilution taken into consideration during calculation of GC concentrations? It should be discussed how it could impact results.
3) Authors should consider including exemplary area under curve area plot in the manuscript
3) The manuscript should be revised and edited, i.e. line 143 contains fragments of the author's guide: "Results This section may be divided by subheadings. It should provide a concise and precise description of the experimental results, their interpretation as well as the experimental conclusions that can be drawn"
Author Response
This manuscript concerns the study association between glucocorticoid diurnal rhythm in breastmilk and infant body composition at age 3 months. Authors obtained different results than previously published, so it is reasonable to further explore this topic. Authors present all the methodologies and results, discussed in details the limitation and advantages of the study and compare obtained results with other work. It would be interesting to compare these results with the data obtained for the infant fed with milk formula as they are not receiving glucocorticoids with milk.
We would like to thank the reviewer for taking his/her time for reviewing our manuscript, as well as for the extensive and useful review comments. Please find our point-by-point rebuttal below.
We also agree with the reviewer with regard to comparing formula-fed vs. breastfed infants, although human milk and infant formula not only differ in glucocorticoid contents. We will keep this in mind when designing a possible follow-up study.
1) Authors could extend the justification of the study in the introduction and discussion and explain the GC rhythmicity and its possible impact on developing an infant.
Thank you for this suggestion. We have added a hypothesis to our introduction (lines 72-74). We expected to find a higher fat mass in infants exposed to a less variable maternal GC rhythm. We have also added a sentence to the first paragraph of the discussion (line 192-193). We hope that this is the justification of the study the reviewer is looking for. If not, please let us know what we could alter further.
2) What was the amount of milk taken for the LC/MS-MS analysis? Was milk dilution taken into consideration during calculation of GC concentrations? It should be discussed how it could impact results.
Thank you for this question. 200μl of milk was necessary for one measurement, and since all samples were measured in duplo, 400μl was necessary for each sample. None of the mothers collected less than this, so no milk dilution was necessary. We have added the necessary amount of milk used in the methods (line 118), as well as added a sentence about the measurements in duplo (line 125). For a more extensive explanation of the laboratory analyses, we would like to refer the reviewer to the publication referenced in our manuscript.[1]
Additionally, we have altered the type of LC-MS/MS analyzer used, since the previous information was outdated (lines 120-121).
3) Authors should consider including exemplary area under curve area plot in the manuscript
For good examples of AUC plots, we would like to refer the reviewer to the article by Pruessner et al.[2] The following graph was produced by these authors to illustrate the AUC ground (left) and increase (right):
To correct for total sample collection time (since these differed between mother-infant pairs), we divided the AUC’s obtained using Pruessner et al.’s method by total collection time in hours.
Since the graphs were produced by other authors, we have decided to not add these to the manuscript. We hope that the reference is illustrative enough for readers. Please let us know if we should add a different example to the manuscript.
3) The manuscript should be revised and edited, i.e. line 143 contains fragments of the author's guide: "Results This section may be divided by subheadings. It should provide a concise and precise description of the experimental results, their interpretation as well as the experimental conclusions that can be drawn"
We have looked in our version of the manuscript, and could not find the example given by the reviewer. We have reviewed the manuscript and altered any typo’s and grammar mistakes we could find.
References
van der Voorn, B.; Martens, F.; Peppelman, N.S.; Rotteveel, J.; Blankenstein, M.A.; Finken, M.J.; Heijboer, A.C. Determination of cortisol and cortisone in human mother's milk. Clin Chim Acta 2015, 444, 154-155, doi:10.1016/j.cca.2015.02.015. Pruessner, J.C.; Kirschbaum, C.; Meinlschmid, G.; Hellhammer, D.H. Two formulas for computation of the area under the curve represent measures of total hormone concentration versus time-dependent change. Psychoneuroendocrinology 2003, 28, 916-931.

Reviewer 2 Report
Clearly state the hypothesis: while the aim of the project is indicated, what is the hypothesis the authors had when they design the project considering that the references cited in the paper ( Hindes and Hahn-Holbrook give opposing findings). Remove the end of line 143 and lines 144, 145 and 146. Line 149: it reads, pairwas. Correct to pair was. Line 150: 3 mother-infant pairs returned the growth questionnaires, but did ( or did not) consent to Pea PD measurements? I suggest that the information on infant sex (n and %) be part of the table (for instance, below gestational age). I would like to see a table with the actual concentration of breastmilk cortisone and cortisol (either in the manuscript or as supplemental data).
Review report: No association between glucocorticoid diurnal rhythm in breastmilk and infant body composition at age 3 months
Objective of the study: The goal of the project was to determine the presence of an association between human milk levels of cortisol and cortisone at 1 month postnatally and growth and body composition of infants at age 3 months taking into consideration the diurnal rhythm of glucocorticoids (GC). While the aim was clearly indicated, there was no identified hypothesis. The proposed question of the study is not novel, but published data is equivocal. The authors tried to address a limitation of previous studies which does not take into consideration milk composition variation over the course of the day.
Introduction:
The authors referred multiple times to the findings of Hahn-Holbrook et al. in which an inverse correlation between milk cortisol and BMIP was found. Hinde and al., on the other hand, found a positive association between glucocorticoids and growth in monkeys. Considering these opposing findings, the authors should describe what their hypothesis is and how the variation of milk GC over the course of the day could prove the listed hypothesis.
Methods:
It is unclear why 3 individuals were removed from the association analysis: in line 150: 3 mother-infant pairs returned the growth questionnaires, but did (or did not) consent to Pea PD measurements? Milk can be fairly variable even within a feeding. Is it known whether the concentration of GC varies from foremilk and hindmilk, such as different nutrients such as fat? If that is the case, did your collection method take that into consideration? Please make the following corrections: Remove the end of line 143 and lines 144, 145 and 146. Line 149: it reads, pairwas. Correct to pair was.
Results:
Although the authors mentioned the sex distribution of the infant population, it would be helpful to indicate it again in Table 1 where the characteristics of the population is describes. It can be dismissed where it is (top of the table). Considering that there are different methods for the measurement of GC, I suggest the addition of a separate table with the actual concentration of breastmilk cortisone and cortisol and their variation over the 24 h. period.
Conclusion:
The conclusion offered the strengths and limitations of the study. Are there any other study that shows a positive correlation between GC and BMI later in life?
Author Response
"Clearly state the hypothesis: while the aim of the project is indicated, what is the hypothesis the authors had when they design the project considering that the references cited in the paper ( Hindes and Hahn-Holbrook give opposing findings). Remove the end of line 143 and lines 144, 145 and 146. Line 149: it reads, pairwas. Correct to pair was. Line 150: 3 mother-infant pairs returned the growth questionnaires, but did ( or did not) consent to Pea PD measurements? I suggest that the information on infant sex (n and %) be part of the table (for instance, below gestational age). I would like to see a table with the actual concentration of breastmilk cortisone and cortisol (either in the manuscript or as supplemental data).
Review report: No association between glucocorticoid diurnal rhythm in breastmilk and infant body composition at age 3 months
Objective of the study: The goal of the project was to determine the presence of an association between human milk levels of cortisol and cortisone at 1 month postnatally and growth and body composition of infants at age 3 months taking into consideration the diurnal rhythm of glucocorticoids (GC). While the aim was clearly indicated, there was no identified hypothesis. The proposed question of the study is not novel, but published data is equivocal. The authors tried to address a limitation of previous studies which does not take into consideration milk composition variation over the course of the day.
We would like to thank the reviewer for taking his/her time for reviewing our manuscript, as well as for the extensive and useful review comments. Please find our point-by-point rebuttal below.
Introduction:
The authors referred multiple times to the findings of Hahn-Holbrook et al. in which an inverse correlation between milk cortisol and BMIP was found. Hinde and al., on the other hand, found a positive association between glucocorticoids and growth in monkeys. Considering these opposing findings, the authors should describe what their hypothesis is and how the variation of milk GC over the course of the day could prove the listed hypothesis.
Thank you for this suggestion. We would first like to address the opposing findings between Hinde et al. and Hahn-Holbrook et al.. At first sight, the findings do seem contradictory. However, Hinde et al. only studied weight gain (in grams per day), without taking length into account. They also studied the Rhesus macaques between 1 and 4 months of age. Hahn-Holbrook et al. studied BMI (a composite variable of weight and length) at the age of 2 years. It might be possible that both weight and length growth is positively associated with milk cortisol, simultaneously leading to a lower BMI growth with a higher cortisol milk content. Alternatively, cortisol in milk might be associated with faster growth early in life, but a more leaner body composition at an older age. This growth pattern is, for example, also seen in breastfed vs. formula-fed infants: breastfed infants have a faster growth in the first few months of life, but their growth is slower at a later age compared to formula-fed infants. We have added a sentence to the introduction to clarify this (lines 57-60)
As for the hypothesis: endogenous GC rhythmicity has been shown to be blunted in both children and adults with obesity. Perhaps exposure to certain maternal GC rhythms early in life (both during pregnancy as well as after birth through breastmilk) might protect against (or increase the risk of) obesity. We have added these considerations to the introduction (lines 72-74).
Methods:
It is unclear why 3 individuals were removed from the association analysis: in line 150: 3 mother-infant pairs returned the growth questionnaires, but did (or did not) consent to Pea PD measurements? Milk can be fairly variable even within a feeding. Is it known whether the concentration of GC varies from foremilk and hindmilk, such as different nutrients such as fat? If that is the case, did your collection method take that into consideration? Please make the following corrections: Remove the end of line 143 and lines 144, 145 and 146. Line 149: it reads, pairwas. Correct to pair was.
Our apologies, we did indeed mean “3 returned the growth questionnaire but did NOT consent to the PeaPod measurements”. We have altered this in the methods (line 161). We have additionally corrected the “pairwas” typo (line 160).
While it is indeed true that milk contents can be fairly variable throughout the day as well as within a feed, a previous study has shown that GC concentrations do not vary between fore- and hindmilk.[1] We have added these considerations to the methods (line 93-95).
Results:
Although the authors mentioned the sex distribution of the infant population, it would be helpful to indicate it again in Table 1 where the characteristics of the population is describes. It can be dismissed where it is (top of the table). Considering that there are different methods for the measurement of GC, I suggest the addition of a separate table with the actual concentration of breastmilk cortisone and cortisol and their variation over the 24 h. period.
We have changed the order of the characteristics described in Table 1; sex is now the fourth characteristic discussed.
We have also added a supplementary table to the manuscript, showing cortisol and cortisone concentrations in the breastmilk of our population in 4-hour intervals (lines 175-176 and Supplementary Table 1).
|
Supplementary Table 1: Cortisol and cortisone concentrations in breastmilk in 4-hour intervals. |
||
|
|
Cortisol (nmol/L) |
Cortisone (nmol/L) |
|
0:00-4:00 |
4.1±5.5 |
15.1±11.8 |
|
4:00-8:00 |
11.6±8.7 |
29.7±12.8 |
|
8:00-12:00 |
8.2±6.5 |
27.9±8.6 |
|
12:00-16:00 |
4.4±2.9 |
21.3±6.8 |
|
16:00-20:00 |
2.1±1.4 |
13.0±6.1 |
|
20:00-24:00 |
2.1±3.9 |
10.4±9.2 |
|
Values represent mean±SD |
||
The following graphs are to further illustrate the GC rhythm in breastmilk (not added to the supplementary files):
Conclusion:
The conclusion offered the strengths and limitations of the study. Are there any other study that shows a positive correlation between GC and BMI later in life?"
Thank you for this question. Most studies in adults show an inverse relation between cortisol and BMI,[2] although another study suggested there might be an interaction with age.[3] However, we wanted to keep the focus on GC rhythmicity, rather than basal GC levels, and have therefore chosen not to reference these studies in our manuscript. We hope reviewer can agree to this. We did, however, add a sentence to the first paragraph of the discussion, summarizing previous findings on the associations between endogenous GC rhythmicity and obesity in both children and adults (line 192-193).

Reviewer 3 Report
The focus of the study is really interesting.
The paper is well written.
I have no observations.
Author Response
The focus of the study is really interesting.
The paper is well written.
I have no observations.
We would like to thank the reviewer for his/her compliments and for taking his/her time for reviewing our manuscript.